Host-Microbe Biology

# Respiratory Bacteria Stabilize and Promote Airborne Transmission of Influenza A Virus

Hannah M. Rowe,[a] Brandi Livingston,[a] Elisa Margolis,[a] Amy Davis,[a] Victoria A. Meliopoulos,[a] Haley Echlin,[a] (ID) Stacey Schultz-Cherry,[a] (ID) Jason W. Rosch[a]

[a]Department of Infectious Diseases, St Jude Children's Research Hospital, Memphis, Tennessee, USA

**ABSTRACT** Influenza A virus (IAV) is a major pathogen of the human respiratory tract, where the virus coexists and interacts with bacterial populations comprising the respiratory tract microbiome. Synergies between IAV and respiratory bacterial pathogens promote enhanced inflammation and disease burden that exacerbate morbidity and mortality. We demonstrate that direct interactions between IAV and encapsulated bacteria commonly found in the respiratory tract promote environmental stability and infectivity of IAV. Antibiotic-mediated depletion of the respiratory bacterial flora abrogated IAV transmission in ferret models, indicating that these virus-bacterium interactions are operative for airborne transmission of IAV. Restoring IAV airborne transmission in antibiotic-treated ferrets by coinfection with *Streptococcus pneumoniae* confirmed a role for specific members of the bacterial respiratory community in promoting IAV transmission. These results implicate a role for the bacterial respiratory flora in promoting airborne transmission of IAV.

**IMPORTANCE** Infection with influenza A virus (IAV), especially when complicated with a secondary bacterial infection, is a leading cause of global mortality and morbidity. Gaining a greater understanding of the transmission dynamics of IAV is important during seasonal IAV epidemics and in the event of a pandemic. Direct bacterium-virus interactions are a recently appreciated aspect of infectious disease biology. Direct interactions between IAV and specific bacterial species of the human upper respiratory tract were found to promote the stability and infectivity of IAV during desiccation stress. Viral environmental stability is an important aspect during transmission, suggesting a potential role for bacterial respiratory communities in IAV transmission. Airborne transmission of IAV was abrogated upon depletion of nasal bacterial flora with topical antibiotics. This defect could be functionally complemented by *S. pneumoniae* coinfection. These data suggest that bacterial coinfection may be an underappreciated aspect of IAV transmission dynamics.

**KEYWORDS** *Streptococcus pneumoniae*, microbiome, transmission

Address correspondence to Stacey Schultz-Cherry, Stacey.Schultz-Cherry@stjude.org, or Jason W. Rosch (requests for materials and data), Jason.Rosch@stjude.org.

Influenza A viruses (IAVs) are major pathogens of birds and mammals. The primary site of human IAV infection is the upper respiratory tract, with more-severe manifestations occurring when the virus accesses the lower respiratory tract. Enhanced IAV morbidity and mortality can also occur due to coinfection with bacterial pathogens, also commonly found in the human upper respiratory tract microbiota. The best-characterized bacterial synergy of IAV is that with *Streptococcus pneumoniae* (1, 2). This synergy operates in multiple aspects of pathogenesis. IAV coinfection enhances transmissibility of *S. pneumoniae* in murine (3) and ferret (4) models. IAV infection and the resultant inflammation can enhance the pneumococcal biofilm-to-plankton transition, enhancing the invasive potential of the pneumococcus (5). In the lung, IAV infection exposes receptors enhancing pneumococcal adherence (6), depletes alveolar macrophages which in turn enhances pneumococcal replication (7), and provides pneumo-

cocci with host sialylated substrates to promote growth (8), and coinfection with both pathogens alters tissue repair processes promoting lethal infection (9). These observations underscore the complex interactions that underlie the synergies between IAV and bacteria during respiratory infection.

It is increasingly apparent that in addition to indirect interactions, direct bacterial viral interactions are also operative, with the most extensive evidence coming from studies focusing on species found in the gastrointestinal tract. Many classes of enteric viruses, including picornaviruses (10), reoviruses (10), and caliciviruses (11), rely on bacteria or bacterial products for infectivity. Such direct interactions have recently been shown to occur also between IAV and respiratory bacteria, including *S. pneumoniae* (12, 13). These direct interactions were shown to enhance the adherence of pneumococcus to cultured respiratory cells *in vitro* and to enhance initial colonization and invasive disease in murine models of otitis and invasive disease (13). These interactions also alter how the host responds to either individual pathogen, as when IAV-pneumococcus complexes were utilized as vaccine antigens, efficacy was greater than that seen with either vaccine alone (12). While the coinfecting bacterial species typically benefit from IAV infection, the roles of bacterial species coinhabiting the respiratory tract in IAV biology remain less well understood. Studies of the respiratory tract microbiome and of susceptibility to IAV infection in household transmission have suggested an important role for the composition of the respiratory tract microbiome in terms of susceptibility to IAV infection (14). Here, we demonstrate that IAV directly benefits from interactions with normal bacterial respiratory flora, with the bacterial partners conferring environmental stability and enhancing airborne transmission of the virus.

## RESULTS

Interactions with bacteria have previously been demonstrated to promote the stability of picornaviruses (15, 16) and reoviruses (16, 17). We hypothesized that bacterial stabilization of environmental IAV via direct interactions may be one mechanism that the virus exploits to retain infectivity following release into the environment. To test this, $10^8$ CFU of washed bacterial cultures representing several respiratory tract-colonizing pathogens, previously shown to associate with influenza A viruses (12, 13), were incubated with $10^{7.5}$ 50% tissue culture infectious doses ($TCID_{50}$) of influenza virus strain A/Puerto Rico/8/1934(H1N1) (PR8) for 30 min, centrifuged, and washed to remove nonadherent virus. The cosedimented material was subjected to desiccation in a SpeedVac and then rehydrated to determine viral infectivity by $TCID_{50}$ assay. SpeedVac-mediated desiccation, while not biologically relevant, allows concentration of bacterial and viral particles and the ability to measure log fold changes in viral viability promoted by the bacterial complex. Real world conditions would subject the bacterium-virus complex to less-harsh desiccation stressors and, furthermore, would occur in the context of host-derived molecules.

Virus desiccated in the presence of *S. pneumoniae* (pneumococcus) or *Moraxella catarrhalis* retained viability and infectivity, whereas IAV complexed to *Staphylococcus aureus*, *Staphylococcus epidermidis*, nontypeable *Haemophilus influenzae*, or *Pseudomonas aeruginosa* did not retain infectivity of IAV (Fig. 1A). The desiccation resistance conferred by the IAV-bacterium complex was independent of bacterial viability, as ethanol-killed pneumococci, or a Δ*spxB* mutant, which maintains higher desiccation viability than wild-type pneumococcus (41), promoted influenza viability to a degree equivalent to that seen with live wild-type *S. pneumoniae* (Fig. 1B). However, the pneumococcus had to be intact to promote infectivity of IAV, as virus cosedimented in the presence of pneumococci that had been killed and lysed with β-lactam antibiotics retained significantly less viability than virus desiccated in the presence of live pneumococci (Fig. 1B). These data indicate that direct interactions between IAV and respiratory bacteria can promote environmental stability of IAV during desiccation in a species-specific manner and that bacterial association alone is not sufficient to stabilize IAV.

The extensively hydrated polysaccharide capsule can promote environmental sur-

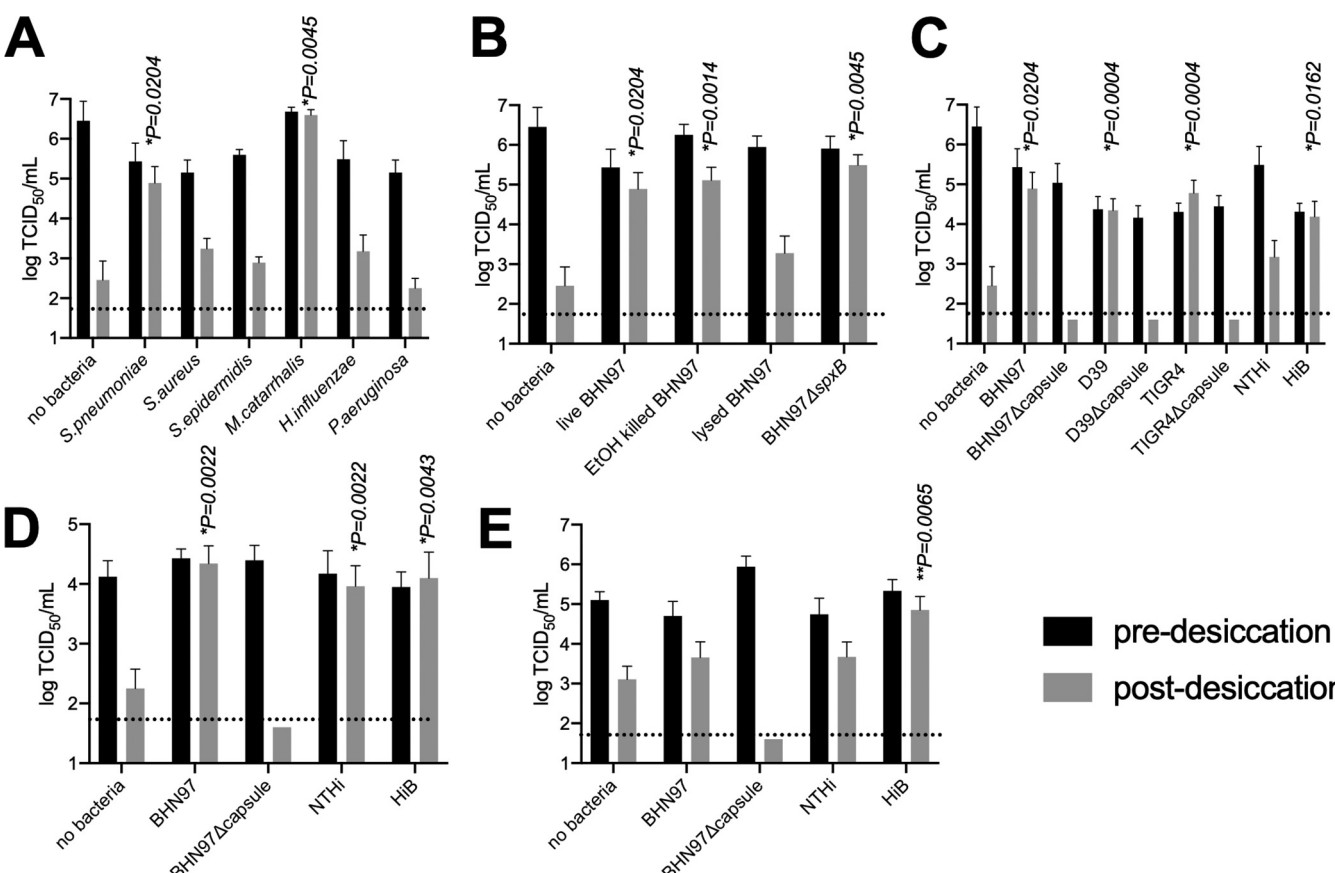

**FIG 1** Nasopharyngeal bacteria promote influenza virus desiccation stability. The indicated bacterial strain was preincubated with influenza virus PR8 (A to C) or A/California/04/2009 (H1N1) (D) or A/Wisconsin/67/2005 (H3N2) (E) followed by centrifugation and washes to remove nonassociated virus and desiccation in a SpeedVac (gray) or was not subjected to desiccation (black). Virus alone was prepared in a small volume (less than 10 µl) of either cell culture media or egg fluid and was directly desiccated in the SpeedVac. (A) Nasopharyngeal tract-colonizing bacteria provide differing degrees of IAV desiccation protection. (B) Pneumococcal viability does not affect desiccation promotion of IAV, as ethanol-killed or ΔspxB pneumococcal mutants with enhanced desiccation tolerance provided protection for IAV infectivity equivalent to that seen with live pneumococci. β-lactam-killed and lysed pneumococci did not promote viability retention. (C) Desiccation survival of IAV with encapsulated and noncapsulated strains of *S. pneumoniae* and *H. influenzae*. (D) Pneumococcal capsule and *H. influenzae* promote stability of A/California/04/2009 (H1N1). (E) *H. influenzae* serogroup B promotes stability of A/Wisconsin/67/2005 (H3N2). Bars represent means and error bars represent standard deviations of results from at least 6 biological replicates. *P* values were calculated by Mann-Whitney testing compared to virus desiccated in the absence of bacteria; the dotted line represents the limit of detection. NTHi, nontypeable *H. influenzae*.

vival of *S. pneumoniae* (18). Further, lyophilization of live attenuated influenza vaccine in sugar-containing solutions promotes maintenance of viability (19). We hypothesized that the polysaccharide capsule may represent one mechanism by which direct interactions between IAV and the pneumococcus promote viral stability. Targeted deletions of the capsule locus were made in three pneumococcal strains, representing three distinct genotypes and serotypes. These mutants, and parental strains, were incubated with IAV strain PR8, cosedimented, and subjected to desiccation. The presence of capsule had no discernible impact on the initial association of IAV with and adherence to the bacterial cells, similarly to previous studies (13), but only virus cosedimented in the presence of encapsulated pneumococcal strains retained infectivity. This phenomenon was not specific to pneumococcal capsule, as IAV cosedimented in the presence of encapsulated *H. influenzae* serotype B strain (HiB) demonstrated significantly enhanced viability compared to that of IAV cosedimented with unencapsulated nontypeable *H. influenzae* (Fig. 1C). These data indicate an important role for the polysaccharide capsule of *S. pneumoniae* and *H. influenzae* in conferring desiccation tolerance to IAV.

To confirm that the findings were also operative in human-relevant pathogens, desiccation experiments were performed with both A/California/04/2009 (H1N1) and A/Wisconsin/67/2005 (H3N2) desiccated in the presence of capsular and nonencapsu-

lated respiratory tract bacteria. Similarly to the results seen with PR8, cosedimentation of A/California/04/2009 with *S. pneumoniae* resulted in significantly enhanced desiccation tolerance, a phenotype that was dependent upon expression of the pneumococcal polysaccharide capsule (Fig. 1D). However, both nontypeable *H. influenzae* and HiB promoted desiccation tolerance of A/California/04/2009, suggesting that the role of *H. influenzae* serotype B capsule is less important in promoting desiccation tolerance of A/California/04/2009 than of PR8 and that another *H. influenzae* surface factor may play a role in stabilizing A/California/04/2009. Interestingly, the desiccation tolerance of A/Wisconsin/67/2005 (Fig. 1E) was not promoted by *S. pneumoniae* regardless of capsule status but was promoted by encapsulated *H. influenzae* serotype B. Taken together, these data suggest that bacteria resident in the human respiratory tract are capable of stabilizing H1N1 strains of IAV but that this stabilization may be subtype specific, with certain respiratory tract bacteria stabilizing certain IAV subtypes.

These observations of respiratory tract-colonizing bacteria conferring desiccation tolerance to IAV suggest a mechanism whereby, during shedding from an infected host, the viral particles associated with specific members of the respiratory tract microbial community may have enhanced environmental stability and, hence, transmissibility. First, to determine if we could alter the respiratory tract microbial community, ferret anterior nasal swabs were collected from 9-week-old castrated male ferrets, and a subset of these animals were treated immediately after sample collection and 3 days later by application of 75 mg mupirocin (Mup) ointment, commonly used in nasal decolonization prior to surgery (20), to the ferret exterior nostrils using a polyester-tipped swab to apply the ointment to the exterior of the nares and interior of the nostrils to the depth of the first turbinate. All ferrets were sampled again 24 h after the final treatment to determine the impact of mupirocin on respiratory bacterial communities. DNA was extracted from swabs, and the V3-V4 region of 16S was sequenced to determine microbial community composition. While total bacterial burden was significantly reduced as determined by the number of 16S rRNA copies per swab (Fig. 2A), the overall community diversity was not significantly altered (see Fig. S1 in the supplemental material). The decrease in bacterial burden was limited to particular species, including multiple Gram-positive cocci and *Moraxella*, which were significantly reduced in abundance in treated but not control animals (Fig. 2B; see also Fig. S2). Relative abundances of the disaggregated taxa are represented in Fig. S3. These data indicate that mupirocin treatment selectively reduced the relative burden of multiple respiratory bacterial species, including *Streptococcus* and *Moraxella*, both of which mediate IAV binding and desiccation tolerance.

On the basis of the observations indicating that specific bacterial species can mediate IAV infectivity during desiccation and that mupirocin depletes bacterial species from the nasal passages, we hypothesized that mupirocin treatment would adversely impact IAV transmission. To determine the effect of this community disruption on airborne transmission of IAV, ferrets were treated by application of mupirocin ointment to the ferret nostrils, on days 1 and 3 prior to viral infection and at each nasal wash collection time point, with ointment administered after collection of nasal wash. Pairs of donor and aerosol-contact ferrets were housed in cages with perforated dividers such that the ferrets could not directly contact each other. Donor ferrets were infected with $10^6$ $TCID_{50}$ A/California/4/2009 (H1N1) in a volume of 1 ml (0.5 ml per nostril) by the intranasal route under isoflurane sedation. Contact ferrets were introduced the following day. Nasal wash samples were collected under conditions of ketamine sedation of the ferrets on days 3, 5, 7, 9, and 11 or 12 after viral challenge from both donor and contact animals to monitor viral burden. Control ferrets with no manipulation of the respiratory tract microbial community had a 75% transmission rate, with two of four ferrets having culturable virus in their nasal washes and an additional contact becoming seropositive by day 21 postchallenge (Fig. 3A and B). Depletion of the respiratory tract microbial community by applying mupirocin to the nostrils of donor and contact ferrets completely abrogated airborne transmission, with no contact animals expressing positive viral titers or seroconverting (Fig. 3C and D) ($P = 0.0221$

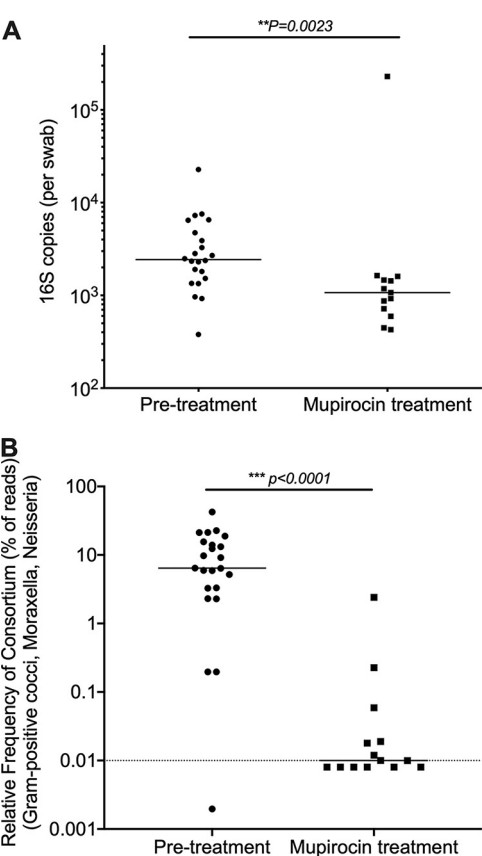

**FIG 2** Impact of mupirocin on ferret respiratory tract microbial community composition. (A) The bacterial content of the nasal passages was significantly lower after mupirocin treatment as measured by the number of bacterial 16S copies recovered on nasal swabs prior to and following treatment. (B) Microbiome content of Gram-positive and Gram-negative cocci (specifically, the relative frequencies of *Moraxella, Neisseria, Lactococcus, Vagococcus, Enterococcus hirae*, "*Streptococcus fryi*," and *Streptococcus suis*) is significantly reduced following treatment. Each dot represents data from a swab collected from an individual ferret. Solid line indicates median for each group, dashed line represents limit of detection. Groups were compared by Mann-Whitney test with a *P* value of <0.05 being considered significant.

compared to no-treatment group). Treatment of only the donors was sufficient to block airborne viral transmission, with no contact animals expressing positive viral titers or seroconverting (Fig. 3E and F). While trending toward significance (*P* = 0.0746), low animal numbers restricted the data from this groups from being statistically significantly different from the data from the no-treatment controls. All of the directly infected animals, regardless of treatment status, shed similar viral loads, seroconverted to infection, and exhibited similar clinical symptoms (see Table S1 in the supplemental material). These data indicate that perturbation of bacterial communities in the respiratory tract of donor ferrets by mupirocin treatment results in reduced airborne transmission of A/California/4/2009 (H1N1) influenza virus.

If our hypothesis that respiratory bacteria promote IAV transmission is correct, then restoring bacterial communities that both bind to and stabilize IAV should rescue IAV transmission in the mupirocin-treated donor animals. Because *S. pneumoniae* was shown to stabilize IAV in the presence of desiccation stress and effectively colonizes the respiratory tract of ferrets, a separate group of mupirocin-treated donor ferrets was colonized with $5 \times 10^6$ CFU in a volume of 0.6 ml (0.3 ml per nostril) mupirocin-resistant *S. pneumoniae* 2 days after IAV challenge, a time when viral shedding is at nearly peak levels (21). The mupirocin treatments used were identical to those described above. Pneumococcal colonization was robust and stable throughout viral sample collection across donor animals (Fig. S4). Viral transmission was restored upon colonization of *S. pneumoniae*, with 60% of contact animals having viable virus in their nasal washes and

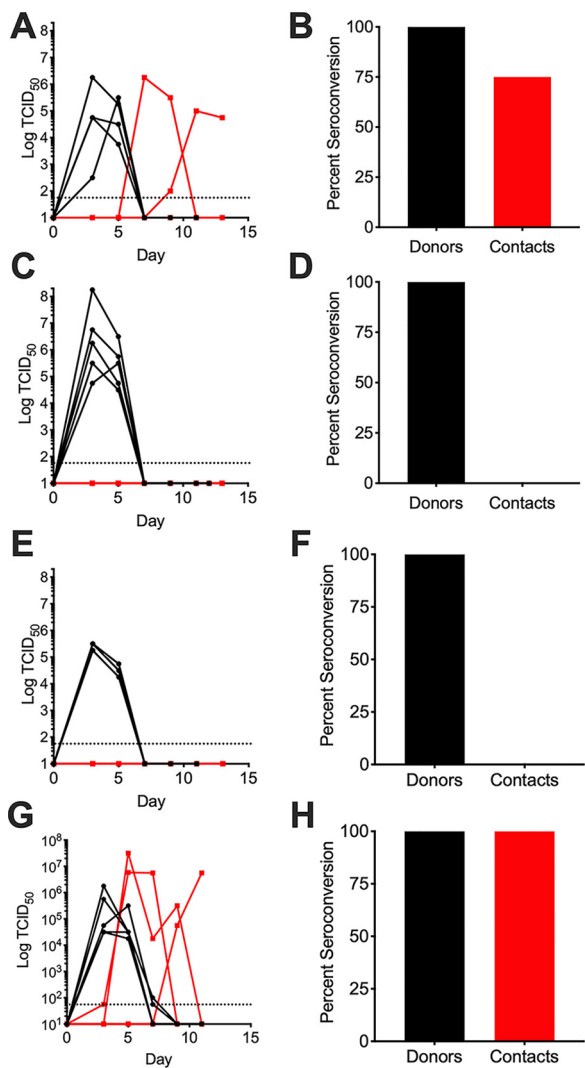

**FIG 3** Nasopharyngeal bacteria promote airborne transmission of influenza virus. Donor ferrets were infected with influenza A virus A/California/04/2009 (H1N1) and paired 24 h postinfection with aerosol-contact ferrets in the same cage with perforated dividers separating the animals. (A, C, E, and G) Influenza virus burden in nasal lavage measured by 50% tissue culture infectious dose ($TCID_{50}$). (B, D, F, and H) Percentage of animals that seroconverted as measured by hemagglutination inhibition (HAI) assay titer greater than 1:80 dilution by day 21 postinfection. Black, donors; red, aerosol contacts. Dotted line represents limit of detection for $TCID_{50}$ assay. Day, day postinfection of donor animals. (A and B) Ferrets with no manipulation of the respiratory tract microbiota; $n = 4$ donors and 4 contacts. (C and D) Nostrils of both donor and contact ferrets were treated with mupirocin ointment; $n = 5$ donors and 5 contacts. (E and F) Donor ferret nostrils were treated with mupirocin ointment; $n = 3$ donors and 3 contacts. (G and H) Donor ferret nostrils were treated with mupirocin followed by colonization with $10^6$ CFU of *S. pneumoniae* strain BHN97Mup$^R$Strep$^R$. Each data point represents results determined for an individual ferret over time.

all contact ferrets seroconverting to infection (Fig. 3G and H). There was no significant difference in contact positivity between the untreated controls and the mupirocin-treated animals recolonized with *S. pneumoniae* ($P = 0.7934$). Comparisons of contact positivity between the *S. pneumoniae*-colonized, mupirocin-treated animals and groups where either the donor alone or both donors and contacts were treated demonstrated enhanced transmission upon bacterial colonization ($P = 0.0149$ and $P = 0.0035$, respectively). While the donor viral loads were similar to those seen with the previous groups, the severity of donor symptoms was higher in colonized ferrets (Table S1), so we cannot rule out the possibility that there was an additional effect resulting from enhanced donor symptoms on shedding to contacts. However, these data suggest that coloni-

zation of donor ferrets by *S. pneumoniae* is sufficient to rescue the IAV transmission defect resulting from mupirocin depletion of the respiratory flora.

## DISCUSSION

Virus-bacterium synergies are inherently complex, with interactions among bacteria, virus, and the host immune system being operative in synergistic as well as antagonistic relationships. We demonstrated that the presence of common members of the human nasopharyngeal microbiome, including *S. pneumoniae*, *M. catarrhalis*, and *H. influenzae*, can enhance desiccation survival of H1N1 influenza A viruses when the virus is in complex with the bacterial surface. In our study, desiccation survival was enhanced by the presence of bacterial capsule in both *S. pneumoniae* and *H. influenzae*, suggesting that the polysaccharide capsule plays an important role in retaining IAV infectivity under these conditions. These findings reflect studies of enteric viruses such as picornaviruses whose stability can be enhanced via direct bacterial interactions (15) or via interaction with bacterial lipopolysaccharide (16). Bacterial lipopolysaccharide and peptidoglycan also enhance thermostability of reoviruses, which in turn promotes infection of host cells following environmental stress (17). Similar results with respect to virion stability and infectivity enhancement have also been observed with human astroviruses (22). This suggests that interactions between viral pathogens and the bacterial communities are likely operative at distinct host niches, including the respiratory tract, in addition to the better-characterized synergies operative for enteric pathogens.

The environmental persistence of IAV is dependent upon multiple factors and can vary considerably between viral subtypes (23). The differing capacities of respiratory bacterial species to promote stabilization of the H1N1 viruses versus the H3N2 subtype suggest that there may be additional important differences between distinct IAV subtypes with respect to their capacity to be stabilized by direct bacterial interactions, with some IAV subtypes requiring distinct bacterial species for binding and stabilization. Further, some IAV strains may not be stabilized by human respiratory tract bacteria but may instead be stabilized by bacteria found in the natural reservoir of the respective IAV strains. Additionally, the respiratory tract microflora of model organisms may impact IAV transmissibility, underscoring the potential importance of the native bacterial flora in investigations of IAV transmissibility. Here, we demonstrated that alterations to the microbial community following mupirocin ointment in IAV-infected donor animals were sufficient to block airborne transmission of H1N1 A/California/4/2009 IAV in a ferret model. Restoration of transmission upon bacterial colonization of the mupirocin-treated donor animals with *S. pneumoniae* further supports this hypothesis. We also cannot rule out the possibility of an impact on microbial community composition and IAV transmission resulting from the presence of the polyethylene glycol PEG ointment carrier alone; however, the restoration of transmission by recolonization with *S. pneumoniae* suggests that whatever effects are present are subtle. It should also be noted that we utilized relatively young ferrets aged 9 weeks, while many other investigations querying influenza transmission routinely utilized ferrets of 4 to 12 months of age (24–26). The utilization of younger ferrets was primarily due to previous work demonstrating that ferrets in this age group rapidly transmit *S. pneumoniae* by both contact and aerosol routes (4, 41). Whether our findings would extend to older ferrets that likely have distinct microbial community composition remains an unresolved but intriguing issue. The relevance of results to various IAV challenge doses is also an important issue, as bacterially mediated synergies may be important only at specific viral thresholds of infectivity.

These findings suggest that, unlike the enteric microflora, which enhances viral infectivity of the same host, the respiratory tract microflora of the infected host is primarily operative in viral infectivity of the subsequent host. In household transmission studies of IAV, *S. pneumoniae* or closely related streptococcal species were identified in approximately 95% of samples collected from both the child index cases and the household contacts who developed influenza (27). These data suggest that modulating

donor respiratory flora via antibiotic exposure or vaccination may profoundly affect IAV transmission. It should be stressed that topical antibiotics were given prior to IAV challenge in our study, with no impact on disease severity in the donor animals. Even in the light of this limitation, targeting bacterially mediated transmission may represent a novel strategy of IAV infection control that could be explored.

## MATERIALS AND METHODS

**Ethics statement.** All experiments involving animals were performed with the approval of and in accordance with the guidelines of the St. Jude Animal Care and Use Committee (St. Jude Children's Research Hospital, Memphis, TN). The St. Jude laboratory animal facilities have been fully accredited by the American Association for Accreditation of Laboratory Animal Care. Laboratory animals were maintained in accordance with the applicable portions of the Animal Welfare Act and the guidelines prescribed in the Department of Health and Human Services (DHHS) publication "Guide for the Care and Use of Laboratory Animals."

**Bacterial and viral strains and growth conditions.** _S. pneumoniae_ strains BHN97 (serotype 19F), D39 (serotype 2), and TIGR4 (serotype 4) were inoculated onto tryptic soy agar (TSA) (GranuCult; Millipore, Burlington, MA) plates supplemented with 3% sheep blood (iTek, St. Paul, MN) and 20 $\mu$g/ml neomycin (Sigma, St. Louis, MO) and then grown overnight at 37°C in a 5% $CO_2$ humidified incubator. The strains were then inoculated directly into Todd-Hewitt (BD, Franklin Lakes, NJ) broth supplemented with 0.2% yeast extract (BD) (ThyB) and grown to log phase for use in experiments. A capsule mutant was generated by transforming SPNY001 genomic DNA containing a Sweet Janus cassette that replaces the capsule locus (28) into strains BHN97, D39, and TIGR4, and the results were confirmed by the lack of latex bead agglutination (Statens Serum Institute, Copenhagen, Denmark) for the respective capsules.

Nontypeable _H. influenzae_ strain 86-028NP (29), originally isolated from a patient with chronic otitis media, and encapsulated _H. influenzae_ serotype b strain 10211 (ATCC) were grown on chocolate agar supplemented with 11,000 units/liter bacitracin (BD) and then directly inoculated into brain heart infusion broth (BD) supplemented with 0.2% yeast extract (BD), 10 $\mu$g/ml hemin, and 10 $\mu$g/ml NAD and grown with aeration to mid-log phase. _Staphylococcus aureus_ strain USA400, _Staphylococcus epidermidis_ strain M23864:W2 (ATCC), _P. aeruginosa_ Xen41 (PerkinElmer), and _Moraxella catarrhalis_ (30) were grown on unsupplemented TSA plates, directly inoculated into brain heart infusion broth supplemented with 0.2% yeast extract, and grown with aeration to mid-log phase for use in experiments. Influenza A virus A/Puerto Rico/8/1934 (PR8) and A/Wisconsin/67/2005 (H3N2) were grown in Madin-Darby canine kidney (MDCK) cells. The A/California/4/2009 virus was grown in allantoic fluid of 10-to-11-day-old embryonated chicken eggs. PR8 is of unknown passage history, A/Wisconsin/67/2005 is from a second cell passage from a third egg passage, and A/California/4/2009 is from a fifth egg passage.

For ferret pneumococcal colonization, _S. pneumoniae_ strain BHN97 was made mupirocin and streptomycin resistant (BHN97 Mup$^R$Strep$^R$) to enable continued treatment of ferrets with mupirocin ointment and collection of nasal lavage with streptomycin to reduce the risk of aspiration pneumonia during ketamine sedation and nasal wash collection. Streptomycin resistance was conferred via mutation of _rpsL_ (TIGR4 Sp_0271) by introduction of a K56T mutation (31) generated by splicing overlap extension (SOE) PCR using two fragments that each had the point mutation. The first PCR fragment amplified 969 bp upstream of and the first 180 bp of _rpsL_ using primers RpsL_Up_F (GCCGTAGTCATCTTTCTTGGCATC)/RpsL_Up_R (CTGAGTTAGGTTTTGTAGGTGTCATTGTTC). The second PCR fragment amplified bp 151 to 414 of rpsL plus 752 bp downstream using primers RpsL_Down_F (GAACAATGACACCTACAAAACCTAACTCAG)/RpsL_Down_R (CTAATTTGAACCCGGGCTAAAGTTAG). The entire SOE PCR product was amplified using RpsL_Up_F/RpsL_Down_R and was transformed into strain BHN97; resistant mutants were selected for on TSA supplemented with 3% sheep blood and 800 $\mu$g/ml streptomycin. Mupirocin resistance was spontaneously generated and selected for by plating a turbid culture of BHN97 _rpsL_$_{K56T}$ on TSA supplemented with 3% sheep blood, 800 $\mu$g/ml streptomycin, and 10 $\mu$g/ml mupirocin and then selecting spontaneously resistant colonies.

**Cosedimentation and desiccation.** Cosedimentation was performed as previously described (13). Briefly, mid-log bacterial cultures were washed and normalized to 10$^8$ CFU/ml in phosphate-buffered saline (PBS). Influenza virus (3 × 10$^7$ TCID$_{50}$ [50% tissue culture infectious dose]) was added, and the samples were rotated for 30 min at 37°C. The samples were centrifuged and washed twice with PBS. Samples not subjected to desiccation were immediately resuspended in 100 $\mu$l 1× penicillin/streptomycin solution (Gibco) and frozen at −80°C for viral quantification. Samples designated for desiccation were spun for 60 min in a SpeedVac until the pellet was dry. Pellets were resuspended in 100 $\mu$l 1× penicillin/streptomycin solution and frozen at −80°C for viral quantification. Viral titers were determined by TCID$_{50}$ on MDCK cells (32). Three to six biological replicates were performed for each strain.

Ethanol-fixed pneumococci were prepared by resuspending 10$^8$ CFU BHN97 in 1 ml of ice-cold 70% ethanol for 5 min on ice. Cells were pelleted, supernatant was removed, and pellets were dried at 55°C for 5 min to remove residual ethanol. Viability loss was confirmed by plating on TSA/blood. $\beta$-Lactam-killed pneumococci were prepared by resuspending 10$^8$ CFU BHN97 in 1 ml 10× penicillin-streptomycin (Gibco) solution–PBS and incubating 30 min at 37°C. Viability loss was confirmed by plating on TSA/blood, and lysis was confirmed by microscopy examination.

**Ferret infection.** All ferrets were maintained in biosafety level 2 (BSL2), specific-pathogen-free facilities. Use of microbiome collection swabs, treatment of nostrils with ointment, infection, and blood collection were conducted under conditions of general anesthesia performed with inhaled isoflurane at 4%. Nasal washes were collected under ketamine sedation following intramuscular injection of ketamine

into the thighs of restrained ferrets. All ferrets were monitored twice daily for symptoms during infection. Weights were measured daily, and temperature data were collected daily from implanted microchips.

Castrated 9-week-old male ferrets (Triple F Farms, Gillet, PA), confirmed to be seronegative for influenza A viruses (seronegativity to influenza B viruses was not tested) prior to start of study, were housed two per cage, separated by a perforated barrier. The experimental groups had 3 to 5 donor-contact pairs. Animals designated for treatment with mupirocin ointment had 75 mg 2% mupirocin–polyethylene glycol (Perrigo) applied to exterior of nostrils and interior of anterior nares up to the first turbinate with a polyester applicator swab (Puritan) 3 days prior to infection, 1 day prior to infection, on the day of infection (postinstillation of virus inoculum), and on each sampling day (postcollection of nasal lavage); untreated animals were not treated at those time points. Donor animals were infected with $10^6$ 50% tissue culture infectious doses ($TCID_{50}$) of influenza A virus A/California/04/2009 (H1N1) mixed in 1 ml of phosphate-buffered saline (PBS), instilled equally between the two nostrils. Contact animals were introduced into the cages, with separation provided by a perforated divider, 24 h after infection of the donors. On day 3, 5, 7, 9, or 11 or 12 postinfection of donor animals, donor and contact ferrets were sedated with ketamine and nasal lavage was collected in 1 ml PBS supplemented with $1\times$ penicillin/streptomycin (Gibco) divided equally between the nostrils. Nasal lavage fluid was stored at $-80°C$ for viral quantification. Animals designated for coinfection with *S. pneumoniae* were treated with mupirocin ointment and infected as described above. Then, on day 2 after influenza virus infection, *S. pneumoniae* strain BHN97 $Mup^R Strep^R$, grown as described in the cosedimentation and desiccation section, was normalized to $5 \times 10^6$ CFU per 600 $\mu$l PBS and instilled equally between the nostrils. Samples were collected as described above, except penicillin was omitted from the PBS solution. Prior to storage at $-80°C$, 30 $\mu$l of nasal wash was removed and serially diluted in PBS and plated on TSA supplemented with 20 $\mu$g/ml neomycin and 3% sheep blood for bacterial quantification. Viral titers were determined by $TCID_{50}$ on MDCK cells (32). Briefly, MDCK cells were infected with 100 $\mu$l 10-fold serial dilutions of sample and incubated at 37°C for 72 h. Following incubation, viral titers were determined by hemagglutination assay (HA) using 0.5% turkey red blood cells and analyzed by the method of Reed and Muench (33). For samples that were negative by HA, residual supernatant was removed and wells were washed once with PBS and then stained for 1 h at room temperature with 0.5% crystal violet in a 4% ethanol solution. Wells were washed with tap water and infected wells determined by destruction of the monolayer. $TCID_{50}$ levels were again determined using the method of Reed and Muench as described above. On days 14 and 21 after IAV challenge, ferrets were sedated with isoflurane, and 1 ml blood was drawn from the jugular vein. Blood was allowed to clot overnight at 4°C. Serum was collected following centrifugation to pellet clot and stored at $-80°C$. For determination of seroconversion, serum was treated with receptor-destroying enzyme (RDE; Hardy Diagnostics) overnight at 37°C. RDE was inactivated via incubation at 56°C for 1 h followed by dilution in PBS for a final dilution of 1:4 and freezing at $-80°C$ for at least 4 h to continue to inactivate neuraminidase. Starting with a 1:40 dilution of sera and serial 2-fold dilutions in PBS, sera were mixed with 4 hemagglutination units of A/California/04/2009 and incubated 30 min at room temperature. Following incubation, equal volumes of 0.5% washed turkey red blood cells mixed with PBS were added to all wells followed by incubation of a further 60 min at 4°C. The hemagglutination inhibition (HAI) titer was read as the most highly diluted well with a negative hemagglutination reaction.

**Microbiome analysis.** Prior to treatment (day $-3$) and again on the day of infection, just prior to the infection, the microbiome was sampled from the anterior nares of all ferrets. A flocked polyester swab (Copan) (flexible minitip) was inserted into the ferret nares to the depth of the first turbinate, and the interior of each nostril was swabbed for 15 s per nostril followed by insertion of the same swab into the other nostril. Swabs were stored dry at $-80°C$ for DNA preparation. DNA was extracted from nasal swabs after resuspension using methods designed to improve capture of bacterial species present in low-abundance samples (34). Bacterial DNA content was assessed using 16S rRNA quantitative PCR with a plasmid containing *Escherichia coli* 16S gene as the standard, previously described primers and probe (35), and Fast Universal PCR Master Mix (TaqMan) supplemented with 3 mM $MgSO_4$. Microbiome 16S rRNA gene amplification was performed using "touch-down" PCR cycling of V3-V4 amplicon (36), and sequencing was performed at the St. Jude Hartwell Center as previously described (37). Classification of reads was done based on phylogenetic placement on the reference tree as follows. Briefly, Illumina MiSeq paired-end reads were run through the DADA2 pipeline (38) (version 1.10.1) to correct sequencing errors and to determine amplicon sequence variants (ASVs). These ASVs were then used to recruit full-length 16S rRNA gene sequences from Ribosomal Database Project release 16.0 (39) to construct a phylogenetic reference data set and tree. The amplicon sequences were then placed onto the reference tree using pplacer (40).

**Statistical analysis.** All tests were performed with GraphPad Prism 7. Comparisons for viral stability and microbial abundance counts were made via Mann-Whitney testing, with a $P$ value of less than 0.05 considered significant. For analyzing time to positivity during transmission, the Mantel-Cox log rank test was used for analysis of data obtained on the first day of viral positivity in contact animals or, in the case of seroconversion without detectable viral burden, animals were considered positive on the last day of the experiment. A $P$ value of $<0.05$ was considered significant.

**Data availability.** Sequence reads are available from NCBI through accession no. PRJNA622834.

## SUPPLEMENTAL MATERIAL

Supplemental material is available online only.

**FIG S1**, PDF file, 0.04 MB.

**FIG S2**, PDF file, 0.1 MB.

**FIG S3**, PDF file, 0.1 MB.
**FIG S4**, PDF file, 0.02 MB.
**TABLE S1**, PDF file, 0.03 MB.

## ACKNOWLEDGMENTS

J.W.R. is supported by 1U01AI124302 and 1RO1AI110618, S.S.-C. by NIAID contract HHSN272201400006C, and all of us by ALSAC. The content is solely our responsibility and does not necessarily represent the official views of the National Institutes of Health.

Author contributions were as follows: H.M.R., B.L., V.A.M., A.D., and H.E. performed the experiments. H.M.R., S.S.-C., and J.W.R. designed the study. H.M.R., E.M., and J.W.R. analyzed the data. H.M.R. and J.W.R. wrote the manuscript, and all of us edited and approved the final manuscript.

We declare that we have no competing interests.

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
