## [Reviewer comments · mSystems]

Respiratory bacteria stabilize and promote airborne transmission of influenza A virus

Hannah Rowe, Brandi Livingston, Elisa Margolis, Amy Davis, Victoria Meliopoulos, Haley Echlin, Stacey Schultz-Cherry, and Jason Rosch

Corresponding Author(s): Jason Rosch, St. Jude Children's Research Hospital

Review Timeline:

Submission Date:	August 4, 2020
Editorial Decision:	August 5, 2020
Revision Received:	August 14, 2020
Accepted:	August 17, 2020

Editor: Jack Gilbert

Reviewer(s): The reviewers have opted to remain anonymous.

Transaction Report:

DOI: <https://doi.org/10.1128/mSystems.00762-20>

August 5, 2020

Dr. Jason W Rosch
St. Jude Children's Research Hospital
262 Danny Thomas Place, Room E8076
Memphis, Tennessee 38105

Re: mSystems00762-20 (Respiratory bacteria stabilize and promote airborne transmission of influenza A virus)

Dear Dr. Jason W Rosch:

After consultation with a few editors, we do have a few minor comments to consider that generally center on requesting that the manuscript be presented more quantitatively, which is appropriate for mSystems:

* As a general comment, it would be appropriate for the results section to quantify statements, especially when presenting data that is subject to a statistical test. For example, p-values should be reported for observations that relate to the data presented in the Figures, even in cases where the claim is that there is no difference between points of comparison.

* L160 should read "microbial community composition"

* While effective as is, the data presented in Figure 2 might be more compelling if the results were shown independently for each of the taxa considered in this analysis. Given that the authors also have estimates on total bacterial load, these data could be presented in a form that impute the total cellular abundance for the specific taxa in question.

To submit your modified manuscript, log onto the eJP submission site at <https://msystems.msubmit.net/cgi-bin/main.plex>. If you cannot remember your password, click the "Can't remember your password?" link and follow the instructions on the screen. Go to Author Tasks and click the appropriate manuscript title to begin the resubmission process. The information that you entered when you first submitted the paper will be displayed. Please update the information as necessary. Provide (1) point-by-point responses to the issues raised by the reviewers as file type "Response to Reviewers," not in your cover letter, and (2) a PDF file that indicates the changes from the original submission (by highlighting or underlining the changes) as file type "Marked Up Manuscript - For Review Only."

Due to the SARS-CoV-2 pandemic, our typical 60 day deadline for revisions will not be applied. I hope that you will be able to submit a revised manuscript soon, but want to reassure you that the journal will be flexible in terms of timing, particularly if experimental revisions are needed. When you are ready to resubmit, please know that our staff and Editors are working remotely and handling submissions without delay. If you do not wish to modify the manuscript and prefer to submit it to another journal, please notify me of your decision immediately so that the manuscript may be formally withdrawn from consideration by mSystems.

To avoid unnecessary delay in publication should your modified manuscript be accepted, it is

important that all elements you upload meet the technical requirements for production. I strongly recommend that you check your digital images using the Rapid Inspector tool at <http://rapidinspector.cadmus.com/RapidInspector/zmw/>.

Sincerely,

Editor

Editor, mSystems

Journals Department
Response to Reviewers

Comment

As a general comment, it would be appropriate for the results section to quantify statements, especially when presenting data that is subject to a statistical test. For example, p-values should be reported for observations that relate to the data presented in the Figures, even in cases where the claim is that there is no difference between points of comparison.

Response

We completely agree. Statistical values for the respective comparisons are now included throughout the manuscript when not indicated otherwise in the respective figures. Additional details with regards to the statistical tests are indicated in the text in the methods.

Comment

L160 should read "microbial community composition"

Response

We agree. Changed as suggested.

Comment:

While effective as is, the data presented in Figure 2 might be more compelling if the results were shown independently for each of the taxa considered in this analysis. Given that the authors also have estimates on total bacterial load, these data could be presented in a form that impute the total cellular abundance for the specific taxa in question

Response

While the reviewer is correct that a quantitative measure of the specific taxa would be preferable for Figure 2 there are two barriers to implementing that in this study/model system. We were surprised to discover that the intra-animal variation in the nasal microbiome was very high. This meant that few animals had more than 1-2 of the Mupirocin impacted consortium in their initial compositions (all animals had at least one) so the results for each taxa independently alone appear too sparse to evaluate. But we agree this information is important for the narrative, and hence we have included this data as a new figure in the supplemental.

Secondly the reviewer suggests multiplying the quantitative total bacterial load (calculated based on quantitative PCR of 16S rRNA DNA amplification) by the semi-quantitative percent composition (calculated based on the number of sequences classified as specific taxa divided by the total number of sequences classified). This is an excellent suggestion but unfortunately in our laboratories we have failed to validate this method (starting with mock communities with known compositions and validating against either cfu or species-specific quantitative PCR assays). We hypothesize that this is due to the denominator of the semi-quantitative percent composition (total number of 16S rRNA amplicons that were capable of being sequenced and classified) experiencing different biases (differing accuracy/length of 16S amplicons influencing sequencing, sampling bias of filtering for high quality/overrepresented sequences, classification biases) that are not consistent across different communities. In ideal circumstances, our preference would be to use indicator species and species/genus specific quantitative PCR directed at those species to demonstrate the impact of an antibiotic. This was impossible due to not having information a priori on the nasal microbiomes of ferrets and the wide intra-animal variation seen in this study.

August 17, 2020

Dr. Jason W Rosch
St. Jude Children's Research Hospital
262 Danny Thomas Place, Room E8076
Memphis, Tennessee 38105

Re: mSystems00762-20R1 (Respiratory bacteria stabilize and promote airborne transmission of influenza A virus)

Dear Dr. Rosch:

Thanks for the edits and rebuttal. I am satisfied that this study is ready for publication.

Your manuscript has been accepted, and I am forwarding it to the ASM Journals Department for publication. For your reference, ASM Journals' address is given below. Before it can be scheduled for publication, your manuscript will be checked by the mSystems senior production editor, Ellie Ghatineh, to make sure that all elements meet the technical requirements for publication. She will contact you if anything needs to be revised before copyediting and production can begin. Otherwise, you will be notified when your proofs are ready to be viewed.

Sincerely,

Jack Gilbert
Editor, mSystems

Journals Department
FigS3: Accept

FigS1: Accept

FigS2: Accept

Table S1: Accept

Fig S4: Accept